# The Evolution of Tissue Engineered Vascular Graft Technologies: From Preclinical Trials to Advancing Patient Care

**Yuichi Matsuzaki** [1], **Kelly John** [1], **Toshihiro Shoji** [1] **and Toshiharu Shinoka** [1,2,3,*]

[1]   Center for Regenerative Medicine, Nationwide Children's Hospital, Columbus, OH 43205, USA;
     Yuichi.matsuzaki@nationwidechildrens.org (Y.M.); John.Kelly@nationwidechildrens.org (K.J.);
     Toshihiro.shoji@nationwidechildrens.org (T.S.)
[2]   Department of Cardiothoracic Surgery, Nationwide Children's Hospital, Columbus, OH 43205, USA
[3]   Department of Surgery, The Ohio State University Wexner Medical Center, Columbus, OH 43210, USA
*   Correspondence: toshiharu.shinoka@nationwidechildrens.org; Tel.: +1-614-355-5732

**Abstract:** Currently available synthetic grafts have contributed to improved outcomes in cardiovascular surgery. However, the implementation of these graft materials at small diameters have demonstrated poor patency, inhibiting their use for coronary artery bypass surgery in adults. Additionally, when applied to a pediatric patient population, they are handicapped by their lack of growth ability. Tissue engineered alternatives could possibly address these limitations by producing biocompatible implants with the ability to repair, remodel, grow, and regenerate. A tissue engineered vascular graft (TEVG) generally consists of a scaffold, seeded cells, and the appropriate environmental cues (i.e., growth factors, physical stimulation) to induce tissue formation. This review critically appraises current state-of-the-art techniques for vascular graft production. We additionally examine current graft shortcomings and future prospects, as they relate to cardiovascular surgery, from two major clinical trials.

**Keywords:** tissue-engineered vascular grafts; biodegradable material; congenital heart surgery; arteriovenous shunts

---

## 1. Introduction

Congenital heart disease is the leading cause of neonatal death and affects almost 1% of all surviving births [1]. Almost a quarter of these patients need extensive reconstructive surgery [2]. In the adult population, cardiovascular diseases (CVD), such as aortic disease, coronary artery disease (CAD), and peripheral artery disease (PAD), are the leading cause of death worldwide [3].

Current synthetic grafts are generally made from non-biodegradable materials such as polytetrafluoroethylene (e-PTFE®) or polyethylene terephthalate (Dacron®) [4]. These grafts can be successful in large diameter surgeries, but in general, synthetic materials are not clinically suitable because they exhibit an increased risk of thrombosis, stenosis, calcification, and infection, while lacking in durability and growth potential [5]. Because synthetic materials are thrombotic in small-diameter blood vessels of 6 mm or less, the saphenous vein, internal thoracic artery, and radial arteries are often used in cases of coronary artery bypass surgery or peripheral artery bypass surgery for obstructive arteriosclerosis below the knee.

Unfortunately, in affected patient populations, there is an insufficient amount of usable native tissue, either for reasons of anatomy or use in previous operations. In order to overcome the limitations associated with autologous and synthetic graft transplantation, the concept of tissue engineering was proposed [6]. Tissue engineering is a scientific field that may solve the problems that plague current

grafts. Tissue engineering is defined as an interdisciplinary field which combines engineering and biomedical principles to create materials that integrate with a patient's native tissue to restore or improve physiological function.

The classical paradigm of tissue engineering includes:

(I)     Cells (i.e., progenitor cells, stem cells);
(II)    Scaffolds (i.e., synthetic, decellularized extracellular matrix);
(III)   Signals (growth factors, chemotactic factors) [6].

The three components are interdependent and essential to each other when attempting to form organized neotissue. Since newly created neotissue is composed of autologous tissue, these constructs would theoretically be thrombo-resistant, less prone to infection, and have growth capacity. In 1986, the first tissue engineered blood vessels were reported by Weinberg and Bell. They consisted of fibroblasts and collagen gel embedded fibers integrated with bovine endothelial cells (EC), smooth muscle cells (SMC), and Dacron mesh [7]. Since then, hundreds of tissue engineered vascular grafts (TEVGs) have been developed and evaluated in animal models but very few have progressed to human clinical trials. In fact, there are only two that are currently undergoing clinical trials in the United States. In 2001, the first clinical trial to evaluate the use of TEVGs in the venous circulation was conducted in children with congenital heart disease by Shinoka et al. [8]. In 2012, a separate clinical study conducted by Niklason et al. began to evaluate arteriovenous shunts for hemodialysis [9].

This review focuses on the traditional role of scaffolds and cells in TEVGs, the status of recent venous and arteriovenous TEVG clinical studies, and discusses their limitations and future prospects.

## 2. Material and Methods

The ideal scaffold is resistant to calcification, stenosis, thrombosis, and infection. From a surgical perspective, it must be easily handled, sutured, and readily available. In addition, it must have sufficient mechanical properties to withstand the hemodynamic changes of its designated system. Initially, the scaffold not only provides a TEVG's structural integrity, but also the structure to which cells attach and remodel [10]. Ultimately, the organized neotissue assumes the mechanical and structural responsibility of a TEVG as the original scaffold deteriorates. In an effort to find the ideal TEVG framework, numerous synthetic and natural materials have been proposed and evaluated. We will briefly detail some of these approaches. (Figure 1)

### 2.1. Biodegradable Synthetic Vascular Grafts

Biodegradable polymers often act as a temporary scaffold of a blood vessel before it is degraded by the body. Degradation of these materials is initially expressed as a loss of mechanical properties, and then followed by a decrease in their mass to volume ratio. The degradation rate of a given polymer depends on their initial molecular weight, exposed surface area, and physical state (11).

Choosing a suitable material is an important first step in scaffold design and depends on various factors, such as immunogenicity, mechanical properties, and rate of degradation.

Four of the more commonly used synthetic biodegradable materials for TEVGs are:

Poly (glycolic acid) (PGA)
Poly (lactic acid) (PLA)
Poly ($\varepsilon$-caprolactone) (PCL)
Poly (glycerol) sebacate (PGS)

The degradation times of these polymers have been reported to be approximately 2 to 6 weeks, 6 to 12 months, 2 to 3 years, and 4 weeks, respectively [11,12]. Below, we briefly describe each individual polymer and their use in preclinical animal studies.

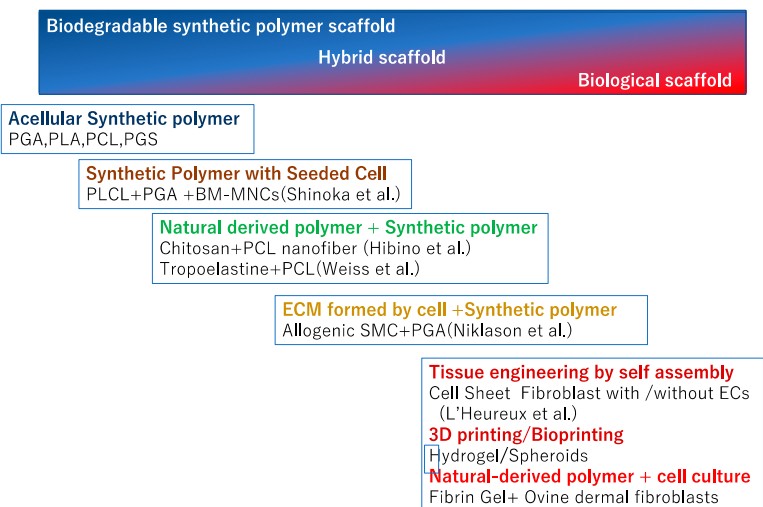

**Figure 1.** Graft classification by scaffold materials in animal model and clinical studies. Various combinations of biological-based scaffolds and biodegradable synthetic-based scaffolds have been developed and evaluated in large animal models and clinical studies. BM-MNCs, bone marrow mononuclear cells; EC, endothelial cell; PCL, poly ($\varepsilon$-caprolactone); PGA, polyglycolic acid; PLA, poly (lactic acid); PGS, poly (glycerol) sebacate; SMC, smooth muscle cells.

### 2.1.1. PGA

Poly (glycolic acid) (PGA) is among the most commonly used polymers in TEVGs. It was one of the first biodegradable polymers to be utilized to fabricate a scaffold. PGA degrades into glycolic acid, which is metabolized by cells and broken down into water and carbon dioxide.

Previous reports have shown that a mouse PGA inferior vena cava (IVC) interposition graft takes up to six weeks to lose its mechanical properties [13].

In an arterial model, a scaffold made of PGA alone does not have sufficient structural characteristics to withstand the arterial blood pressure. Furthermore, the rate of PGA degradation does not allow for adequate neotissue to form, often resulting in aneurysmal dilatation as the PGA begins to lose its mechanical integrity. However, the mechanical properties of PGA grafts can be improved through smooth muscle cell seeding or by being combined with other polymers [14].

PGA can be synthesized singly or as a copolymer such as PLA (PLGA). The strength, hydrophilicity, and degradation rate can be adjusted by changing the lactide: glycolide ratio in the PLGA copolymer or the D:L ratio in the lactide monomer [15]. PGA/PLGA degrade more quickly than PLA alone, and the ease with which the degradation time can be adjusted makes them clinically attractive. For this reason they are used as absorbable sutures and orthopedic implants [15,16].

### 2.1.2. PLA

Poly (L-lactic acid) (PLA) is another biodegradable polymer widely used in tissue engineering. PLA is more hydrophobic than PGA due to the presence of an extra methyl group. Their lower affinity to water leads to longer degradation times (6–12 months) [14]. Thus, studies have reported that PLA takes from months to years to lose its mechanical properties in vivo.

As a result of its hydrophobic characteristics, it displays a tendency towards early thrombogenicity. Therefore, researchers have tried to reduce PLA's thrombotic tendency through cell seeding and/or making chemical surface modifications. Hashi et al. investigated an electrospun, biodegradable PLA TEVG in the arterial circulation of a rat aorta [13]. They suggested that graft patency could be improved by seeding bone marrow mesenchymal cells. The same group saw a 25% improvement in patency when a PLA graft containing hirudin, a thrombin inhibiting polypeptide, was implanted in the abdominal aorta of rats [17].

Additionally, most biodegradable synthetic arterial TEVGs use electrospun nanofibers to withstand the high pressures of the arterial system. Specifically, we have shown electrospun PLA nanofiber grafts have good mechanical properties with high patency in small animal models [18].

Unfortunately, thin fibers and small pore electrospun nanofibrous scaffolds inhibit cell migration which leads to delayed neotissue formation. In contrast, thick fiber and large pore electrospun grafts can enhance the neovascularization and remodeling process by mediating macrophage polarization towards the M2 phenotype [19]. However, electrospun grafts with large pores cause significant blood leakage and exhibit premature loss of mechanical properties. Despite success in small animal models, our current application of these designs to the higher pressures of large animal models have been prone to thrombus formation and graft rupture.

### 2.1.3. PCL

Poly (ε-caprolactone) (PCL) is an aliphatic polyester that has been thoroughly investigated. PCL has one of the slowest degradation profiles out of all biodegradable polymers [14]. PCL is hydrophobic and exhibits a low melting point. Therefore, the PCL polymer is pliable at lower temperatures, in addition to being highly tunable and easy to fabricate [20].

A series of in vivo studies have evaluated electrospun PCL in rats at 3, 6, and 9 month timepoints. The PCL graft patency was 100% and did not display signs of thrombosis. PCL scaffolds were also found to lose 20%, 50%, and 78% of their initial molecular weight at 3, 12, and 18 month timepoints [21,22].

As a result of it's long durability and ease of handling, PCL is frequently used to create grafts suitable for arterial environments [23]. PCL demonstrates favorable mechanical properties with 70% elongation at break compared to 30% and 25% for PGA and PLA, respectively [14]. Additionally, while numerous studies on a multitude of PCL grafts have been conducted in mice and rats, the material has yet to be fully characterized because rodent lifespans expire before the material fully degrades.

In the case of Poly(l-lactic-co-ε-caprolactone)(PLCL), controlling the composition ratios and molecular weights of the individual polymers from which it is derived, allows for better control of mechanical properties and degradation rates. For example, the PLCL (LA/CL=75/25) heteropolymer is stronger than PCL individually, but manages to keep the 70% elongation at break of PCL [14]. We have confirmed the in vivo feasibility of PLA–PLCL scaffolds in high pressure, small-diameter arterial environments [24]. However, when we attempted to translate the material to a sheep carotid artery model, all grafts ruptured and thrombosed. PCL is a hydrophobic material, and without modifications, will most likely lead to a thrombotic or occluded graft.

### 2.1.4. PGS

Poly(glycerol sebacate) (PGS) is a soft (~300 kPa) thermoset elastomer [25] and is believed to support constructive remodeling through its rapid absorption and material properties. In vitro studies have shown that 20% of PGS mass is lost within 30 days, whereas there is 70% loss in vivo [26].

Wu et al. demonstrated rapid remodeling and good patency with PGS grafts implanted in rat arteries [26]. The grafts showed complete endothelialization and stained positive for α-SMA and major histocompatibility complex (MHC). Rats survived up to 12 months, with all postoperative failures being related to acute obstruction [27]. The neoartery showed native compliance. Perivascular innervation was observed and response to vasoactive drugs was limited. Notably, the amount of insoluble elastin was comparable to that of native arteries. Quantifying elastin is difficult, and the studies that have attempted to do so remain limited. Nevertheless, with respect to TEVGs, this remains the largest amount of elastin production reported to date. While PGS is a potentially exciting material, there are no reports that it has been successful in large animal models, most likely because of its rapid degradability and lack of strength.

## 2.2. Natural Polymeric Biomaterials

Conceptually, hybrid scaffolds are bio-based materials that enhance biocompatibility and cellular infiltration by incorporating components of a native extracellular matrix (collagen, gelatin, elastin, fibrin). In addition, naturally occurring polymers such as chitosan and silk can improve the overall structural integrity of a graft. Many biological-based scaffold materials have been investigated in small animal models [28,29].

Scaffolds fabricated from naturally occurring polymers offer the potential of improved biocompatibility through enhanced cellular interactions and reduced likelihood of a foreign body reaction. Nevertheless, the biodegradation profiles and mechanical properties of these polymers can be less than optimal. Therefore, natural biomaterials are often combined with biodegradable synthetic polymers or are chemically modified. Of particular note to tissue engineering are elastin and chitosan.

Elastin is a primary constituent of the extracellular matrix and confers compliance to vessels, thereby ensuring smooth blood flow by the storing and releasing of elastic energy. Elastin itself is highly insoluble and a difficult material to handle when creating scaffolds [30]. Hence, there are few scaffolds composed of elastin. Instead of creating an entire scaffold made out of elastin, it is used primarily to supplement a scaffold composed of another material. Weiss et al. have reported successful results in a rabbit model with a two-layer scaffold consisting of an inner layer of recombinant human tropoelastin and an outer layer of PCL [31]. These grafts show similar mechanical strength to a human intrathoracic artery and display low thrombogenicity. The crosslinking of tropoelastin is paramount to increasing its mechanical strength, and these grafts appear promising, however further research is needed to evaluate their scaffold characteristics.

Chitosan (CS) is a natural polymer obtained from shellfish, and in addition to the ease with which it can be chemically modified, it shows good strength, a porous structure, and antibacterial characteristics. CS has been successfully used in bone, cartilage, and skin tissue engineering constructs [32].

Hibino et al. conducted a study on an electrospun PCL-chitosan nanofiber TEVG in a sheep carotid artery model which showed patency of 67% (*n* = 6) [33]. When integrated within a scaffold, rapidly degrading chitosan is thought to promote better cell invasion and neovascular remodeling due to their large pore sizes.

Another technique used to incorporate the strength of naturally occurring extracellular matrix (ECM) into a scaffold is tissue decellularization. Decellularization has been evaluated in a multitude of studies and involves removing most of the cellular and antigenic components of a tissue. The decellularized tissue would then theoretically leave an intact ECM with preserved mechanical properties. A heterologous TEVG constructed with small intestinal submucosa has been successfully transplanted into a sheep model with good results [34]. In this study, the group seeded with endothelial cells (ECs) showed good patency, but the unseeded control group occluded within 15 days. These studies indicate that decellularized vascular scaffolds will result in thrombosis unless the scaffold is endothelialized or receives additional modifications.

In addition, ECM elements can be exposed to physical and chemical stress during a decellularization process, which can adversely affect the biomechanical properties of an ECM. The disadvantages of decellularized materials include the inability to alter the content and structure of an ECM, variability between donor sources, and risk of viral transmission from animal tissues.

The most sophisticated decellularized approach has been successfully implemented by Niklason et al. Their human decellularized TEVG is prepared in vitro using allogeneic human vascular

SMCs [9]. This TEVG has undergone a multicenter clinical trial and will be further described in detail in Section 4.

## 2.3. Application of Cell Biology to TEVG

Endothelial cells and smooth muscle cells are the main components of the intima and media of a blood vessel, respectively. They were incorporated in some of the first tissue engineered graft designs given their importance in vascular function. ECs, SMCs, and fibroblasts are all essential for creating a stable intima. In addition, SMCs account for the majority of an ECM and ultimately define the mechanical properties of scaffolds. Therefore, early TEVG studies enthusiastically studied EC and SMC populations. Early TEVG studies have shown that seeding mononuclear cells on biodegradable grafts promotes rapid intima formation [35] and demonstrated physiological properties comparable to human blood vessels [36]. However, in order to avoid neointimal hyperplasia, hyperproliferation of SMCs must be controlled.

ECs are responsible for many physiological functions and the synthesis of many important regulators and growth factors [37]. It is extremely important to establish a confluent EC monolayer on the luminal surface of a TEVG to confer resistance to hyperplasia and thrombosis. There is a report that transplantation of EC-seeded ePTFE grafts resulted in a significantly higher patency rate compared to an uninoculated ePTFE control [38]. Interestingly, in another study looking at the use of Dacron seeded grafts, it has been reported that ECs in the neointima of the conduit functions at less than 10% of the physiological level found in natural vasculature [39]. Furthermore, it has been reported that 95% ECs seeded onto a graft is lost within 24 hours [39]. Although the limited number of ECs in the lumen of a TEVG may provide beneficial resistance to neointimal hyperplasia and seems to prevent acute thrombosis, the role that EC seeding has on TEVGs should be further investigated in the future.

L'Heureux et al. pioneered the tissue engineering by self assembly approach (TESA), where they cultured sheets of autologous fibroblasts, and fused these sheets via dehydration around a stainless steel mandrel [40]. Subsequently, cultured autologous ECs would be seeded onto the lumen of the scaffold followed by more culturing in a bioreactor. The TESA approach showed promising functional results in early clinical trials as the first ten patients that underwent arteriovenous shunt graft transplantation displayed a primary patency rate of 78% at 1 month (7/9) and 60% at 6 months (5/8) [41].

However, in addition to a manufacturing cost of greater than $15,000 per graft, another criticism of this approach is that it will not help patients who need rapid intervention because it can take up to 9 months to manufacture [42].

## 2.4. Application of 3D Printing TEVG

3D printing is a field that is currently undergoing rapid development and may serve to advance tissue engineering by incorporating non-uniform characteristics that match patient anatomy and increase the ease fabrication [43,44]. The 3D printing TEVG approach enables the creation of linear or branched tubular structures using synthetic or natural biomaterials [45,46]. The main techniques used for 3D printing of biological materials are ink jet, micro extrusion, and laser assisted printing [47,48]. Within these techniques, hydrogels are the most commonly used material because of their liquid-like state and ease of chemical tunability [49,50]. However, to date, hydrogels do not have the needed initial mechanical properties required of a TEVG [51]. Therefore hydrogels are often supplemented with other biomaterials or cell seeding.

3D printed grafts can also forgo the use of hydrogels and use synthetic and/or natural materials. In fact, Hibino et al. produced an electrospun graft made out of PCL and chitosan [52]. In a sheep model, a custom mandrel was 3D printed based on preoperative image processing. Afterwards, the scaffold created by electrospinning PCL and chitosan fibers around this custom mandrel showed satisfactory remodeling 6 months post-operation.

Bioprinting is an extension of the 3D printing field and utilizes cells and/or other media to print tissue- and organ-like structures [53,54]. As individual cells are difficult to work with, they are often aggregated into "spheroids" [55]. 3D bioprinted structures rely on the self assembling nature of spheroids to fuse and create viable neotissue [56] Itoh et al. reported the use of bioprinting multicellular spheroids and transplanting the resulting scaffold into rat abdominal aortas (*n* = 5) [57]. The TEVGs remained patent until the five day end point of the study, but longer term evaluations are needed to further validate this paradigm.

3D bioprinting as a field has improved immensely in the last decade through the use of spheroids and the improvement of new manufacturing and bioink technologies. While this promising field will likely become more prevalent in tissue engineering applications, focus must be paid to improving ink and scaffold mechanical properties before translation to clinical studies.

## 3. Clinical Study in Venous Model

In 2001, Shinoka et al. reported the successful transplantation of a TEVG which replaced a 2 cm segment of a four-year-old girl's pulmonary artery [58]. The initial conduit was seeded with cells originally from an explanted peripheral vein. These cells were expanded for eight weeks in culture, then subsequently seeded onto a tubular biodegradable scaffold made of a 50:50 copolymer of $\varepsilon$-polycaprolactone–polylactic acid and reinforced with woven poly (glycolic acid) (PGA) fibers. The graft was fabricated in conjunction with the Japanese textile company GUNZE LTD. The PLCL sponge layer was selected because of its biodegradability and pore size which allow cell infiltration and neotissue formation. The material properties of the PLCL allow for flexibility and ease of handling, however its suture retention strength was limited. A woven layer of PLA or PGA was selected to reinforce the graft as it could be easily incorporated into the manufacturing process of the PLCL sponge layer (Figure 2a–d). The patient had no postoperative complications and displayed a patent graft upon follow-up. After the successful procedure and two additional proof of concept large animal studies, we began a pilot clinical trial investigating the use of biodegradable TEVGs in congenital heart surgery.

Between September 2001 and December 2004, 25 Japanese patients underwent extracardiac total cavopulmonary connection (TCPC) operations with a TEVG [59]. Patient demographics, diagnosis, graft type, and size at the time of implantation are presented in Table 1. The median patient age at the time of operation and mean follow-up time was 5.5 and 11.1 years, respectively [60,61]. There were no incidences of graft-rupture, aneurysmal formation, or ectopic calcification. While eight patients died within the follow-up period, none of the mortalities were due to graft-related complications. In fact, the autopsy of one patient who died 13 years after TEVG implantation revealed a structure similar to that of a native vein or pulmonary artery (Figure 3) [62]. However, seven patients presented with asymptomatic graft stenosis and underwent successful balloon angioplasty, including one that required a repeat catheterization and stent placement. Additionally, one patient's TEVG had a thrombus formation one year after implantation, but this was successfully resolved with anticoagulation drug therapy. When contacted, all surviving patients reported no functional limitations (Figure 4) [61].

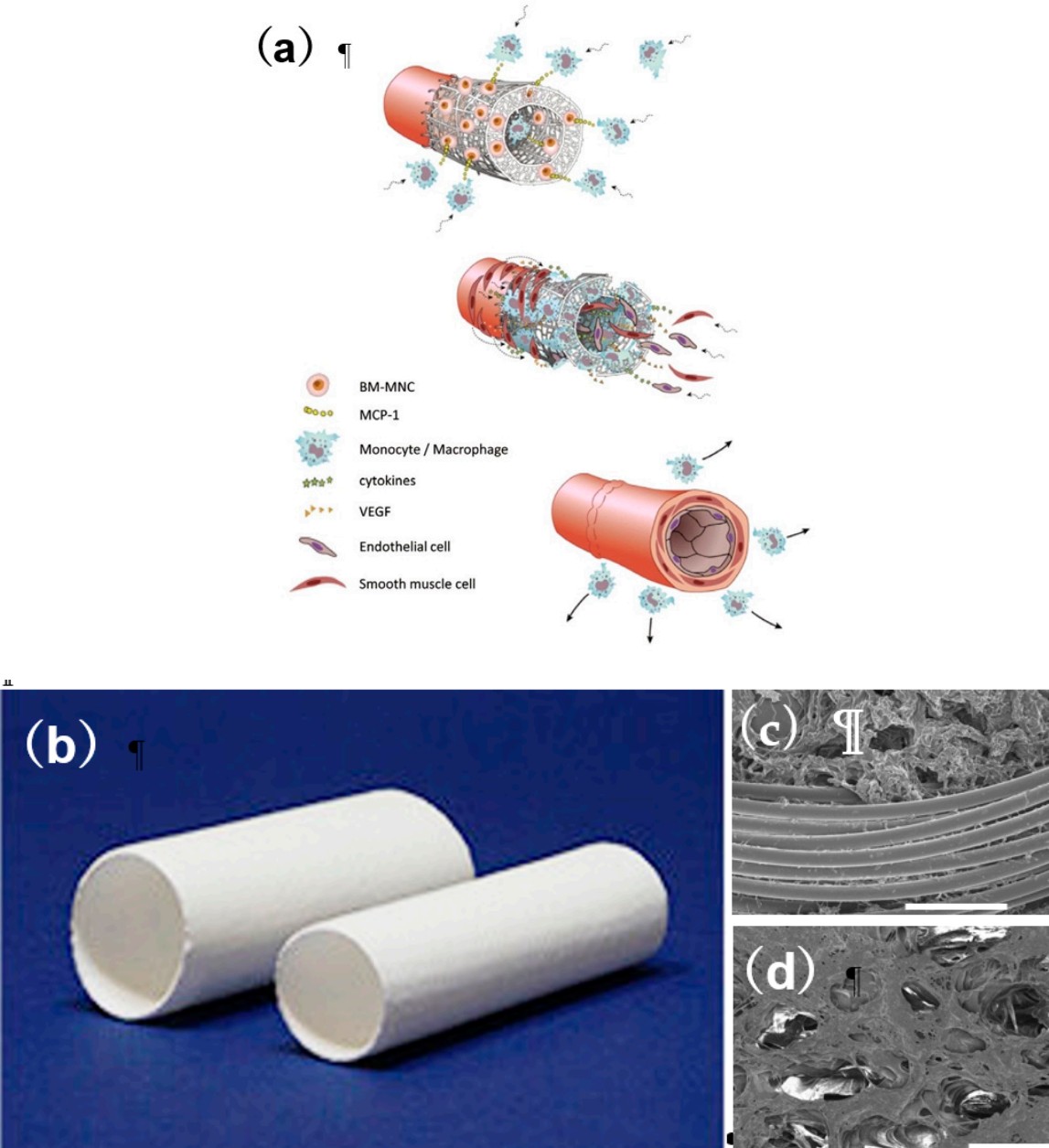

**Figure 2.** (**a**) Proposed mechanism of neovessel formation after implantation of a cell-seeded biodegradable scaffold. Early pulse of monocyte chemoattractant protein-1 (MCP-1) and related cytokines from seeded bone marrow-derived mononuclear cells (BM-MNCs) enhances early monocyte recruitment to the scaffold. Infiltrating monocytes release multiple angiogenic cytokines and growth factors (e.g., vascular endothelial growth factor (VEGF)), which recruit smooth muscle cells. (**b**) Appearance of tissue engineered vascular graft (TEVG). TEVG has a thickness of 1 mm, and mainly has a length of 13 cm; they are trimmed and implanted according to the anatomy of the patient. (**c**) SEM image of Woven poly (glycolic acid) (PGA). (**d**) SEM image of luminal PCLA; the bar shows 100 μm adopted with permission from Shinoka et al.

**Table 1.** Characteristics of the patients.

| Patient | Main Diagnosis | Age | Graft Type | Graft Size |
|---|---|---|---|---|
| 1 | Asplenia, AVSD(A), small RV | 2 | PLA | 16 |
| 2 | Asplenia, SRV, TAPVC(Ib+III) | 1 | PLA | 20 |
| 3 | Concordant criss-cross heart, DORV, PAA, MS | 8 | PLA | 18 |
| 4 | TA(Ib) | 22 | PLA | 24 |
| 5 | PPA, ASD(II), sinusoidal communication | 13 | PLA | 22 |
| 6 | SRV, DORV, AVVA | 4 | PLA | 20 |
| 7 | Total sinus defect, ASD, TR(IV) | 14 | PLA | 24 |
| 8 | Asplenia, SLV, CAVVR | 17 | PLA | 24 |
| 9 | TA(Ib) | 22 | PLA | 22 |
| 10 | Polysplenia, SRV | 4 | PLA | 12 |
| 11 | HLHS, MA, IAA(A) | 2 | PLA | 16 |
| 12 | Asplenia, SRV, PAA, non-confluent PA | 2 | PGA | 16 |
| 13 | SLV, lt AVVA | 2 | PGA | 16 |
| 14 | DORV, small LV, VSD, PS, ASD(II) | 2 | PGA | 18 |
| 15 | polysplenia, cAVSD, DORV, PS | 2 | PGA | 12 |
| 16 | Asplenia, SRV, CA, TAPVC(Ib) | 2 | PGA | 16 |
| 17 | PPA, RA thrombosis, AFL, af | 24 | PGA | 18 |
| 18 | SRV, DIRV, PA, ASD(II) | 1 | PGA | 16 |
| 19 | Asplenia, SRV, PS, CA | 11 | PGA | 18 |
| 20 | polysplenia, cAVSD, PS, CAVVR | 2 | PGA | 14 |
| 21 | DORV, VSD, small RV, PLSVC, TAPVC(IIb) | 3 | PGA | 16 |
| 22 | PPA, ASD(II), PS, Sinusoidal communication | 5 | PGA | 18 |
| 23 | SLV, DILV, PA, ASD, bilateral SVC | 4 | PGA | 18 |
| 24 | Asplenia, SRV | 13 | PGA | 16 |
| 25 | TA(IIc), SAS | 2 | PGA | 18 |

af, atrial fibrillation; AFL, atrial flutter; ASD, atrial septal defect; AVSD, atrio-ventricular septal defect; AVVA, atrioventricular valve atresia; CA, common atrium; cAVSD, complete atrioventricular septal defect; CAVVR, common atrioventricular valve regurgitation; CAVV, common atrioventricular valve; DILV, double-inlet left ventricle; DIRV, double-inlet right ventricle; DORV, double-outlet right ventricle; HLHS, hypoplastic left heart syndrome; IAA, interruption of aortic arch; LV, left ventricle; MA, mitral atresia; MS, mitral stenosis; PA, pulmonary artery; PAA, pulmonary artery atresia; PLA, poly (lactide acid); PLSVC, persistent left superior vena cava; PPA, pure pulmonary atresia; PS, pulmonary stenosis; RA, right atrium; RV, right ventricle; SAS, subaortic stenosis, SLV, single left ventricle; SRV, single right ventricle; SVC, superior vena cava; TA, tricuspid atresia; TAPVC, total anomalous pulmonary venous connection; TR, tricuspid regurgitation; VSD, ventricular septal defect.

In our next steps, we focused on elucidating the cellular and molecular mechanism resulting in neotissue formation and graft stenosis. Surprisingly, in a murine model, we found that that the cells seeded onto the graft all but disappeared within 24 hrs and were replaced by an abundant macrophage response [63,64]. To characterize the effect of the macrophages, we ran experiments that knocked out macrophage function, only to find that the grafts failed to remodel into neotissue. On the other hand, in unseeded grafts, too much macrophage infiltration was found to lead to stenosis [63,64]. While the original tissue engineering paradigms believed that cells seeded onto a scaffold would eventually constitute the neotissue, we found this not to be the case. We found that host-macrophage infiltration is essential for vascular neotissue formation, while excessive infiltration underlies stenosis. Our findings suggested that achieving the proper balance of macrophage infiltration and function is critical for successful neovessel formation [65].

Subsequently, we have continued to investigate the effect of seeding bone marrow-derived mononuclear cells on TEVGs in lambs, as the ovine model is preferred by the FDA due to its propensity to undergo accelerated calcification. As previously mentioned, when we began our original large animal studies in the late 1990s, we expanded cells from an explanted vein, a process that would take anywhere from 8 to 12 weeks [66]. Cell expansion is a time-consuming and costly process. Additionally, cultured cells are susceptible to infection. Being able to utilize bone marrow as a cell source cut a process that required two separate surgeries over the course of a few months to one procedure that

could be completed in a day. We are developing a closed, disposable system which will allow for improved seeding efficiency and reduce the risk of infection [67].

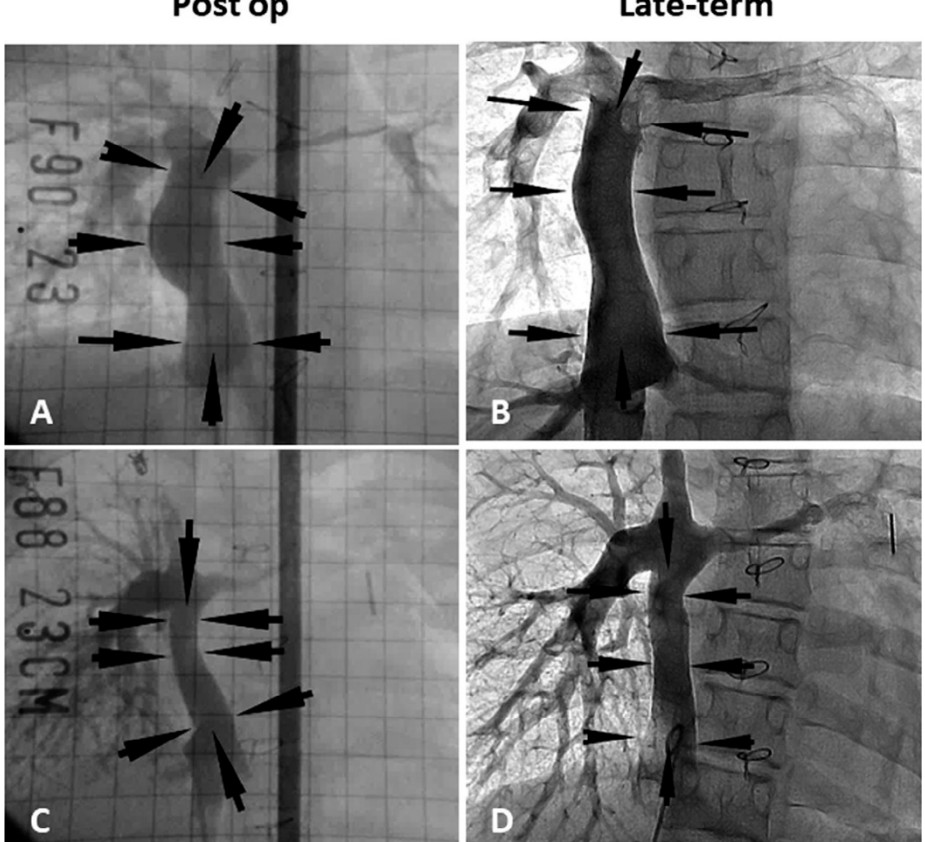

**Figure 3.** Postoperative and late-term TEVG angiography. (**A**) Postoperative angiography in patient 22. (**B**) Angiography 11 years after implantation in patient 22. (**C**) Postoperative angiography in patient 23. (**D**) Angiography eight years after implantation in patient 23. Arrows indicate the TEVG location. Angiography shows macroscopic growth of the TEVGs. Adopted with permission from Shinoka et al.

In regards to the development of graft stenosis, our focus is aimed at modulating the macrophage response. To date, we have shown that pharmacological treatment with a Tgfbr1 inhibitor prevents macrophage induced stenosis and future studies will look at the ability of losartan, an angiotensin II receptor blocker, to mitigate the formation of graft stenosis [68].

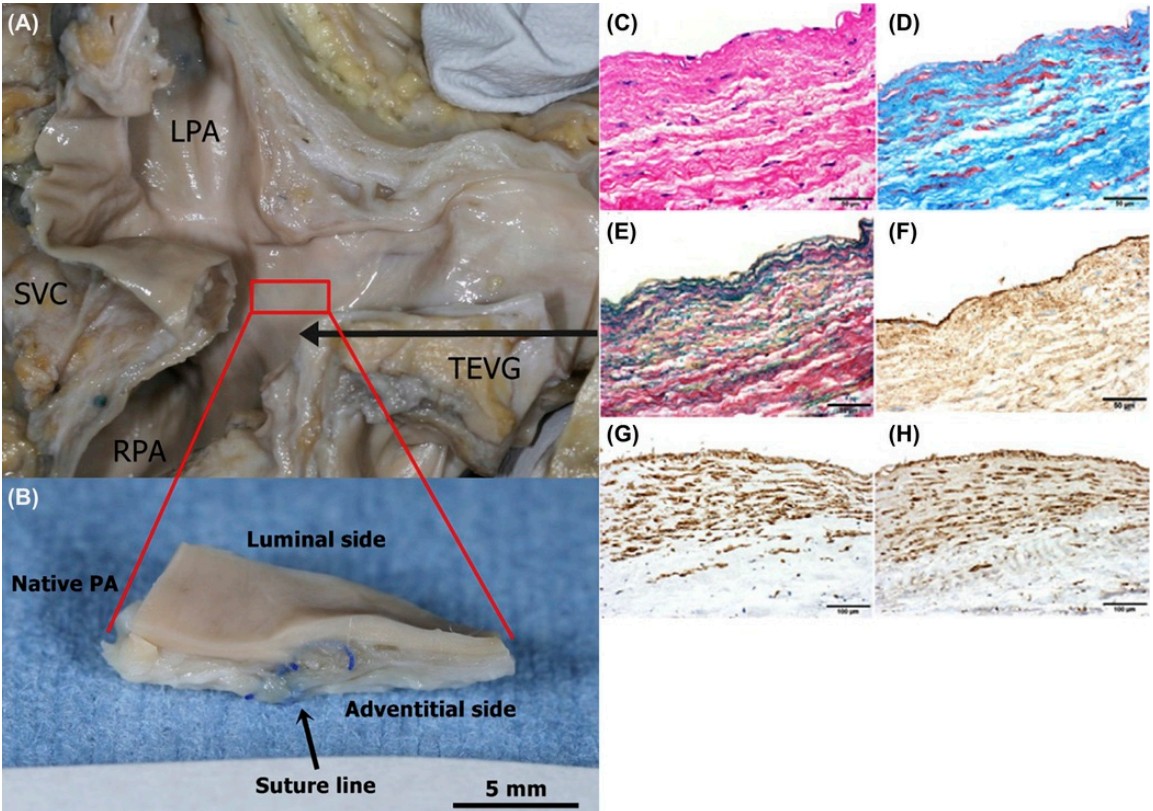

**Figure 4.** Macroscopic and histological images of explanted human TEVG. A female patient with a single right ventricle underwent TCPC with common atrioventricular valve plasty at age 4 years. A 12-mm-diameter biodegradable synthetic TEVG with BM-MNCs was implanted between the hepatic vein and pulmonary artery. Unfortunately, she died 12 years later and an autopsy was performed. (**A**) Macroscopic imaging of the TEVG. No difference appears between native PA and TEVG. (**B**) Cross-section view of the border between native PA and TEVG. (**C**) Hematoxylin-eosin staining. (**D**) Masson trichrome staining. (**E**) Victoria blue-van Gieson staining. (**F**) Factor VIII–positive cells (brown). (**G**) α-Smooth muscle actin–positive cells (brown). (**H**) Calponin-positive cells (brown). LPA, left pulmonary artery; PA, pulmonary artery; RPA, right pulmonary artery; SVC, superior vena cava; TCPC, total cavopulmonary connection; TEVG, tissue engineered vascular graft. Modified from Matsumura G, Shinoka T. First report of histological evaluation of human tissue-engineered vasculature. J Biotechnol Biomater 2015; 5:200 Adopted with permission from Shinoka et al.

## 4. Clinical Study in Arterial Model

Historically, the development of TEVGs in the arterial system has faced many difficulties as the currently available materials have been unable to withstand high arterial pressures while simultaneously allowing for graft degradation and host remodeling. The human acellular TEVG for dialysis access in patients with End-Stage Renal Disease ( is a promising new technology that has been developed by Niklason et al. of [9]. In their approach organ donor-derived SMCs are seeded onto PGA scaffolds and then cultured in a pulsatile bioreactor to stimulate extracellular matrix production. The ester hydrolysis which occurs on the PGA fiber surface results in increased hydrophilicity with an increase in the adsorption of serum proteins and contributes to SMC adhesion. After the initial SMC seeding period of 30 minutes, the bioreactor was filled with medium and the SMCs were cultured for eight weeks under conditions of beating radial stress. The histological examination suggests that pulsatile stimulation is required to promote smooth muscle cell migration. [69]. After the successful creation of an ECM rich scaffold, the grafts were decellularized to remove allogeneic antigens and implanted as vascular scaffolds within baboons and canines to evaluate their in vivo function and remodeling over time.

Their first set of animal experiments evaluated a small diameter (3–4 mm) TEVG in a canine peripheral arterial and coronary bypass model. The graft was created as detailed above with the requirement that cells were sourced from a canine in order to avoid cellular rejection after implantation. It should be noted that following the decellularization process they additionally seeded the grafts with ECs over an additional two days [69]. After excluding for surgical mortality in the setting of patent grafts, they demonstrated long-term patency in five out of six grafts at one month [69].

A baboon model preclinical evaluation was undertaken to evaluate the graft as an arteriovenous shunt. Briefly, grafts 6 mm in diameter were implanted as arteriovenous shunts between an axillary artery and brachial vein. A total of eight grafts were implanted, followed by routine ultrasound to determine patency and evaluated at explant of one, three, or six months. Patency rates were 100%, 66%, and 100% at one, two, and six month explants, respectively. Only one graft showed thrombosis at three months, likely because of technical difficulties with access, which required prolonged manual pressure that led to graft clotting. Hence, the sum total patency of the arteriovenous 6 mm TEVGs in the baboon was 88% (seven of eight) [70]. Following these encouraging preclinical results, human clinical trials were conducted. Two phase two clinical trials were conducted in a total of 60 patients with the graft implanted as an arteriovenous shunt. Mean follow up was 16 months with 12 month patency rates of 28% primary patency, 38% primary assisted patency, and 89% secondary patency [9]. Secondary patency was defined as functional access patency. These cell-free transplants did not elicit an immune response and no aneurysms were reported in the study. An acellular TEVG partially biopsied at 16 weeks showed re-population of host SMC, fibroblasts, and luminal ECs. This host cell repopulation appears to progress over time, as another specimen obtained at 55 weeks showed further progressive remodeling with SMA more highly expressed throughout the vessel [9].

## 5. Conclusions

In recent years, we have seen the adaptation of tissue engineering techniques from the bench to the bedside to advance the field of cardiovascular biology and improve patient care. The methods of vascular tissue engineering are diverse and seek to capitalize on advances in material design and cellular biology. Clinical trials have demonstrated innovative applications of tissue engineering principles but highlight the limitations of the currently employed techniques. The ideal TEVG should be readily available, cost effective, easy to handle, and offer growth potential if required. Furthermore, the scaffold should be resistant to the development of thrombosis, stenosis, calcification, or infection as neotissue assumes physiologic vascular function.

Further research will focus on optimizing scaffolds to improve the prospects of current technologies and advance the field of cardiovascular surgery. The difficulty in scaling graft technologies from small to large animal models within the arterial circulation underscores the challenges that remain before the more widespread adoption of these innovations to patients. Despite these difficulties, the clinical translation of TEVGs is greatly anticipated based on market forces and patient needs.

**Author Contributions:** (I) Conception and design: All authors; (II) Administrative support: All authors; (III) Provision of study materials or patients: All authors; (IV) Collection and assembly of data: All authors; (V) Data analysis and interpretation: All authors; (VI) Manuscript writing: All authors; (VII) Final approval of manuscript: All authors.

**Funding:** This research was funded by Grant from Gunze Limited (Tokyo Japan).

**Acknowledgments:** Matsuzaki was the recipient of a Funding Award from the Uehara Memorial Foundation 2019 (Tokyo, Japan).

**Conflicts of Interest:** Shinoka has received grant support from Gunze Ltd. Other authors reports no conflicts of interest.

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
