# Peer review of "The Evolution of Tissue Engineered Vascular Graft Technologies: From Preclinical Trials to Advancing Patient Care"

_applsci, doi:10.3390/app9071274_

Reviewer 1 Report

This review manuscript reported/examined the currently available biomaterials used for vascular grafts. Authors additionally examined the current problems and future prospects, as they relate to cardiovascular surgery, from two major clinical trials.

The manuscript is clearly structured in Introduction, Analysis, Conclusion and Bibliography. The text is comprehensible and concentrates on the essential results. The abstract presents the essential statements of the article in a logical context.

This paper is worth of publication in the Journal of Applied Sciences, if the following changes/revisions can be applied/addressed:

-        Authors should clearly emphasize the main contributions of this research in the introduction section. More recent studies done by other researchers should be introduced/discussed.

-        In both introduction and conclusion, the authors need to clarify their results clearly. Adding a comparison table could be useful in this regard. Authors main contributions in this review study is not clear yet.

-        Experimental setup is clear; however, in Figure 1, thickness and dimensions should be added. More real photos or microscopy images of the manufactured devices can be added.

-        The results have been clearly shown; however, the manuscript suffers from lack of sufficient discussion for the presented results.

-        Authors should explain about the stats methods used in the manuscript (for comparison); Also, number of samples/tests and errors from each study are missed.

Author Response

Reviewer 1

Comments and Suggestions for Authors

This review manuscript reported/examined the currently available biomaterials used for vascular grafts. Authors additionally examined the current problems and future prospects, as they relate to cardiovascular surgery, from two major clinical trials.

The manuscript is clearly structured in Introduction, Analysis, Conclusion and Bibliography. The text is comprehensible and concentrates on the essential results. The abstract presents the essential statements of the article in a logical context.

)This paper is worth of publication in the Journal of Applied Sciences, if the following changes/revisions can be applied/addressed

-        Authors should clearly emphasize the main contributions of this research in the introduction section. More recent studies done by other researchers should be introduced/discussed.

Thank you for the suggestion.  The introduction section has been modified and sections 2, 3 and 4 supplemented with more recent studies and complete explanation.(3P 66- 10P 342)

-        In both introduction and conclusion, the authors need to clarify their results clearly. Adding a comparison table could be useful in this regard. Authors main contributions in this review study is not clear yet.

Thank you for your comment. The results have been organized with greater clarity. Tables and lists were created to highlight the most commonly used degradable polymer templates in vascular tissue engineering.(12P Fig2) 

- Experimental setup is clear; however, in Figure 1, thickness and dimensions should be added. More real photos or microscopy images of the manufactured devices can be added.

Thank you for your comment. The information for figure 1 has been updated.(11P Fig1)

-        The results have been clearly shown; however, the manuscript suffers from lack of sufficient discussion for the presented results.

Thank you for the valuable comment. The results of the preclinical studies have been further reviewed, specifically as it related to sample number, patency rates and histologic evaluation. Additionally, the human clinical trial data have been further clarified with regards to rates of patency and histologic findings.(8p257-10P356)

-        Authors should explain about the stats methods used in the manuscript (for comparison); Also, number of samples/tests and errors from each study are missed.

Thank you for the suggestion. We supplemented study results by incorporating sample number and time to follow up. We did not reanalysis any results for statistically accuracy but sought to report only those results which were significant.

Reviewer 2 Report

The paper entitled "Biocompatible tissue engineered vascular grafts for cardiothoracic surgery" shows different strategies of tissue engineering for the production of suitable vascular grafts. The review highlights different in vitro and in vivo studies that have been conducted for the production of biocompatible materials and their applications in some clinical venous and arterial models.

Authors could improve the quality of the manuscript by elucidating how integrated approach of recent progress in 3D imaging, 3D printing and tissue engineering technology could promote neovessel formation. Could current bioprinting efforts reduce the steps necessary to construct a scaffold? Please also add references related to this topic. 

Author Response

The paper entitled "Biocompatible tissue engineered vascular grafts for cardiothoracic surgery" shows different strategies of tissue engineering for the production of suitable vascular grafts. The review highlights different in vitro and in vivo studies that have been conducted for the production of biocompatible materials and their applications in some clinical venous and arterial models. 

Authors could improve the quality of the manuscript by elucidating how integrated approach of recent progress in 3D imaging, 3D printing and tissue engineering technology could promote neovessel formation. Could current bioprinting efforts reduce the steps necessary to construct a scaffold? Please also add references related to this topic.  

Thank you for this valuable consideration.  We have added an additional section detailing the most commonly employed 3D printing techniques and highlighted recent advances and publications related to 3D printing in vascular tissue engineering. We have additionally cited recent relevant reviews related to this matter.(7p231-8p256)